

# Spatial effects of trade, foreign direct investment (FDI), and natural resource rents on carbon productivity in the GCC region

Haider Mahmood

Department of Finance, College of Business Administration, Prince Sattam bin Abdulaziz University, Saudi Arabia

## ABSTRACT

**Background:** Natural resource rents (NRRs) may determine the environment and economic growth of the GCC countries due to their over-reliance on the natural resource sector. NRRs are the source of income in resource-abundant GCC countries. So, increasing income of these countries could pollute the environment by increasing overall economic activities. Consequently, NRRs could determine carbon productivity in the GCC region through increasing income and carbon emissions.
**Methods:** The effects of trade openness (TO), foreign direct investment (FDI), urbanization, and oil and natural gas rents on carbon productivity (CP) are examined in the GCC region from 1980–2021 using the spatial Durbin model.
**Results:** The CP of the GCC countries has spillovers in their neighboring countries. Oil rent reduces carbon productivity in domestic economies and the entire GCC region. Natural gas rent, TO, and FDI increase, and urbanization reduces carbon productivity in neighboring economies and the entire GCC region. Moreover, urbanization reduces carbon productivity in domestic economies as well. The study recommends the GCC countries to reduce reliance on oil rent and increase globalization in terms of TO and FDI in the region to promote carbon productivity. Moreover, GCC countries should also focus more on natural gas rent instead of oil rent to raise carbon productivity.

# INTRODUCTION

The 21[st] Conference of Parties (COP21) targets to reduce greenhouse gas emissions and decrease 1.5–2 centigrade global warming (*Murshed et al., 2022*). Similarly, COP 26 extends to adopt emission-mitigation policies, and the Sustainable Development Goals (SDGs) also target sustainable growth and development worldwide (*Hak, Janouskova & Moldan, 2016*). To achieve the SDGs, economic growth must be achieved with less carbon emissions, which can be achieved by raising carbon productivity (CP). *Kaya & Yokobori (1997)* demarcate it as the national income divided by carbon dioxide ($CO_2$) emissions. So, CP is the gross domestic product (GDP) per unit of $CO_2$ emissions. Increasing CP does

Corresponding author
Haider Mahmood,
haidermahmood@hotmail.com

mean that an economy has more production per unit of $CO_2$ emissions. This definition also reflects that an economy may reduce $CO_2$ emissions with a given level of output. Thus, it is not focusing only on reducing absolute $CO_2$ emissions, but it is aiming to increase income per unit of emissions. Therefore, carbon productivity may be considered a strong indicator of sustainable growth. An economy can increase carbon productivity by reducing energy intensity, raising energy efficiency, reducing the consumption of fossil fuels, and increasing cleaner energy consumption. Therefore, GDP may be increased with less carbon emissions and carbon productivity can be enhanced resultantly. *Beinhocker et al. (2008)* claimed that global $CO_2$ emissions could be reduced by enhancing carbon productivity. Furthermore, we cannot ignore the importance of economic growth to target lower carbon emissions in an economy. Thus, economic growth should be targeted by increasing income per unit of emissions. In this way, sustainable growth can be achieved by increasing carbon productivity in the economy to achieve SDGs.

Trade can be considered a major driver of carbon productivity (GDP/$CO_2$ emissions). For instance, the difference between exports and imports is a component of the definition of the GDP. If exports are more than imports, then the surplus trade balance can increase the economic growth of an economy. Thus, surplus trade could help to raise GDP, a nominator of the ratio of CP. On the denominator of CP, trade may create a scale effect on carbon emissions by increasing economic activities and energy consumption (*Grossman & Krueger, 1994*; *Mahmood et al., 2023*). Moreover, Trade Openness (TO) might be responsible for reducing ecological standards in a country. Low ecological standards may be established to decrease the cost of exporting firms to achieve competitiveness in the international market (*Revesz, 1992*). Thus, trade openness can increase $CO_2$ emissions and may reduce carbon productivity. In contrast, trade may have technological and knowledge spillovers in an economy and could foster research and development (R&D) activities (*Grossman & Helpman, 1990*). The resultant technological innovations may increase carbon productivity. Moreover, trade raises economic growth and demand for a cleaner environment, which would shift an economy from dirty industries to clean industries and processes (*Arrow et al., 1995*). Thus, trade could have technique and composition effects. Moreover, trade might motivate economies to follow the strong ecological standards of developed countries (*Birdsall & Wheeler, 1993*). Therefore, trade can affect both GDP and carbon emissions, which could determine carbon productivity.

Foreign direct investment (FDI) inflows could also affect both GDP and carbon emissions. For instance, FDI increases economic activities and thus supports economic growth. Foreign investments are filling a gap between saving and investment in any country and are increasing additional economic activities other than local investments. Thus, FDI is adding to the GDP of the economy. On the environmental side, foreign investors bring the latest technologies to the FDI recipient economies. Thus, FDI would become a source of clean technology transfer and would help in reducing carbon emissions (*Letchumanan & Kodama, 2000*). Technology transfer in an economy can also help raise energy efficiency, which could reduce emissions (*Dessus & Bussolo, 1998*). Thus, FDI could play a positive environmental role by reducing $CO_2$ emissions and would raise carbon productivity. On the whole, FDI could help increase CP by increasing GDP and reducing

$CO_2$ emissions. On the contrary, FDI can be responsible for increasing emissions if FDI comes in a pollution-oriented industry due to low environmental standards (*Cole, 2004*; *Mani & Wheeler, 1998*). Thus, FDI may reduce carbon productivity. With any positive or negative environmental contribution, FDI and trade openness are potential drivers of carbon emissions and can determine carbon productivity.

Natural resource rents (NRRs) might raise the GDP of natural resource-abundant nations as NRRs are the significant source of income in resource-rich nations. However, NRRs can be a cause of unsustainable growth in these economies. Because natural resources are limited on the earth and their over-extraction would limit the future growth of resource-rich economies (*Meadows et al., 1972*). For instance, the Dutch Disease elucidates that rising natural resource exports may obstruct the growth of other economic activities. For instance, the discovery and export of natural resources may give a big push to total exports. As a result, the increasing overall exports may appreciate the local currency, which makes non-resource exports expensive in the international market. Consequently, exports from the non-resource sector would decrease. Moreover, imports become cheaper because of the appreciation of local currency, which might increase total imports. Thus, increasing imports may increase the trade deficit and would reduce the GDP of the country. In this way, the overall GDP of the resource-rich nation may decline due to natural resource exports (*Sachs & Warner, 1995*). Along the same line, many recent studies found a negative impact of NRRs on GDP growth (*Sha, 2023*; *Ze et al., 2023*; *Safdar, Khan & Andlib, 2022*; *Hordofa et al., 2022*). However, some recent literature has also corroborated the positive effect of NRRs on GDP growth (*Ben-Salha, Dachraoui & Sebri, 2021*; *Islam, Al Abbasi & Dey, 2023*). Furthermore, *Yu (2023)* has reported mixed effects of NRRs on GDP growth in the alternative investigated countries. Thus, the literature substantiates that NRRs are a potential determinant of economic growth in resource-rich nations. Regarding the environmental aspect of carbon productivity, NRRs can increase emissions (*Peng et al., 2023*; *Danish, Ulucak & Baloch, 2023*; *Umar et al., 2020*). Nevertheless, *Hodžić, Šikić & Dogan (2023)* found that NNRs decreased emissions. Therefore, NRRs can determine carbon emissions, economic growth, and carbon productivity.

The GCC countries are resource-rich nations. In particular, the oil sector is more dominant in these economies. The oil sector could have environmental concerns. A gas mixed with hydrocarbons is heavily utilized in the extraction of oil, which is released directly into the air after performing its function in oil extraction. In addition, oil and gas extractions release massive amounts of combustion gases, including carbon emissions and hydrocarbons, which are responsible for public health and environmental health (*Johnston, Lim & Roh, 2019*; *Lange & Redlinger, 2019*). Therefore, oil and gas extractions are highly polluted processes and can be responsible for higher $CO_2$ emissions, which can reduce carbon productivity in GCC countries. However, oil and gas rents are the significant percentage of GDP in the GCC countries supporting their economic growth. To tackle the environmental problem, the GCC countries cannot sacrifice their GDP as the incomes of these economies are heavily reliant on the oil and gas sectors. Therefore, the GCC economies should focus on the increasing production of oil and natural gas per unit

of $CO_2$ emissions instead of reducing absolute emissions. Taking into account the expected negative environmental effects and positive economic growth effects of the oil and gas sectors, NRRs would impact CP in GCC countries. Hence, the present study investigates the effects of oil and gas rents on CP instead of $CO_2$ emissions.

Realizing the prominence of NRRs in determining the environment, *Mahmood & Furqan (2021)* investigated the effect of Oil Rent (OR), and *Majeed et al. (2021)* studied the effect of total NRRs on $CO_2$ emissions in GCC countries. Furthermore, the literature has investigated the effect of FDI (*Zmami & Ben-Salha, 2020*; *Al-Mulali & Tang, 2013*; *Mahmood & Furqan, 2021*) and the impact of urbanization (*Mahmood & Furqan, 2021*; *Zmami & Ben-Salha, 2020*) on $CO_2$ emissions. Nevertheless, these researches are dedicated to the causes of absolute $CO_2$ emissions. Nevertheless, a study on sustainable economic growth in terms of carbon productivity is missing in the GCC literature. In addition, the GCC region is heavily dependent on natural resources and cannot target a reduction in $CO_2$ emissions by cutting the production of the oil and gas sectors. However, the GCC region can target sustainable economic growth by increasing economic outputs per unit of $CO_2$ emissions in both resource and non-resource sectors. Therefore, the effects of OR and Natural Gas Rent (NGR) should be tested on carbon productivity in these countries. Nevertheless, a study on testing the impact of NRRs on CP is missing in both the GCC and the global literature. Moreover, carbon productivity could have spatial dimensions in the GCC countries due to their trading agreements and common landscape. Thus, the present study aims to test the effects of OR, NGR, urbanization, FDI, and TO on CP using spatial econometrics in geographically nearby GCC countries. These analyses would guide policymakers to achieve sustainable growth in this region.

## LITERATURE REVIEW

CP is pertinent for sustainable growth in a country (*Murshed et al., 2022*). The literature explored different determinants of CP. *Wang, Li & Li (2022)* explored 114 economies from 2000–2014 and reported that renewable energy consumption (REC) increased, and non-REC decreased carbon productivity. Furthermore, urbanization and income inequality played a mediator role in reducing carbon productivity. *Song & Han (2022)* explored 30 Chinese provinces from 2006–2018 and found that ecological standards increased carbon productivity. For instance, strong ecological standards may force to reduce $CO_2$ emissions. Hence, the reduced $CO_2$ emissions might increase carbon productivity. *Tian & Yang (2020)* investigated 31 Chinese industries from 2003–2015 and corroborated that oil prices reduced carbon emissions. Hence, increasing oil prices helped to raise carbon productivity in an oil-importer economy. *Li & Liu (2022)* examined the total factor carbon productivity (TFCP) and originated that technological efficiency and progress helped increase TFCP in Central Asia from 1991–2019. For instance, technological progress might improve the total factor productivity (TFP) by increasing production per unit of all types of inputs. Thus, it helps to increase TFCP as well due to increasing production per unit of $CO_2$ emissions. In addition, a nonlinear impact of urbanization on TFCP was discovered. Thus, the initial urbanization increased the TFCP. However, urbanization reduced TFCP after a threshold point.

*Zhang & Xu (2016)* investigated 35 industries in China from 2006–2014 and found that environmental regulations and innovations helped develop technologies that increased carbon productivity in labor-intensive industries. Moreover, FDI helped to improve CP, and the industrial structure reduced CP. *Xie, Li & Zhang (2022)* investigated the Economic Belt of the Yangtze River and found that reducing pollution-oriented industrial clusters improved carbon productivity. Furthermore, large-scale clusters of green-based industries also improved carbon productivity. *Bai et al. (2019)* examined the TFCP in 88 economies from 1975–2013 and found that economic growth and R&D helped in convergence to a high TFCP club. However, the energy intensity and trade were responsible for the convergence of the countries to a low TFCP club. *Li & Wang (2019)* examined and corroborated that income level, technological progress, and trade openness accelerated carbon productivity in China.

*Meng & Niu (2012)* investigated China and found that technological innovations and optimized structure of exports increased China's carbon productivity. *Su, Chen & Lin (2023)* examined all provinces of China from 2006–2017 and corroborated that innovations in wind and solar technologies helped to raise carbon productivity. Particularly, these technologies replacing coal consumption showed a greater effect on carbon productivity. *Zhou, Huang & Zhang (2023)* investigated the role of Innovative City Pilot (ICP) on carbon productivity in Chinese cities from 2003–2016 and found that ICP enhanced carbon productivity. Particularly, innovation, industrial advancement, and energy efficiency also contributed to carbon productivity. *He et al. (2023)* examined the effect of agglomeration on CP in 27 Chinese cities from 2006–2020 and found that industrial agglomeration and technological progress improved CP.

*Jahanger, Usman & Ahmad (2022)* examined the impact of globalization on CP in 30 provinces of China from 2009–2017. The authors reported that globalization accelerated carbon productivity after a threshold point and concluded a non-linear relationship. *Geng, Sun & Wang (2022)* studied and reported that the carbon-trading policy raised carbon productivity in 30 Chinese provinces. *Murshed et al. (2022)* investigated the E-7 economies from 2007–2018 and found that globalization and energy efficiency increased, and urbanization, financial inclusion, and income level decreased carbon productivity. However, urbanization, financial inclusion, and globalization interacting with energy efficiency increased carbon productivity. *Sai, Lin & Liu (2023)* investigated 35 African countries from 2005–2017 and found that technical efficiency improved carbon productivity. However, green finance and financial agglomeration showed a negative effect on carbon productivity. Moreover, this impact was reduced with increasing development levels and increased with the use of fossil fuels.

Carbon emissions are global emissions and would have spatial links. Considering this argument, many studies are conducted on spatial dimensions to examine the determinants of carbon productivity. *Long, Shao & Chen (2016)* applied spatial econometrics to 30 provinces in China from 2005–2012 and found a significant spatial correlation in CP. It means that the CP of a province is helping to raise carbon productivity in neighboring provinces with a spillover effect. Furthermore, energy efficiency and technology helped to improve carbon productivity. However, economic growth and the industrial energy mix

reduced CP. *Yao, Zhang & Zheng (2022)* explored the spatial effect of green credit in 30 provinces in China from 2003–2016 and found that green credit enhanced local and neighboring carbon productivity. Moreover, carbon productivity showed spillovers in neighboring provinces. *Zhang et al. (2018)* investigated the Chinese CP in spatial analyses of 30 provinces from 2000–2014. They found that carbon productivity had spillovers among provinces and exports and imports also showed positive spillovers on carbon productivity. In particular, imports played a more significant role in raising carbon productivity. *Liu & Zhang (2021)* investigated 30 Chinese provinces from 1998–2017 in spatial analysis and found a nonlinear impact of industrial clusters on carbon productivity. Moreover, technological innovations helped to shape this relationship. *Meng, Sun & Guo (2022)* investigated 30 Chinese provinces from 2011–2020 and found significant spatial spillovers in carbon productivity. Thus, the carbon productivity of a province raised carbon productivity in neighboring provinces. Furthermore, REC increased carbon productivity.

*Han (2021)* probed 30 Chinese provinces from 2009–2017 and found that technological innovations helped to raise carbon productivity in direct and spillover estimates. Additionally, the spatial links of carbon productivity were also found to be significant between provinces. Hence, the CP of a province helped to increase CP in neighboring provinces. *Tang et al. (2022)* examined 25 Chinese provinces from 2007–2019 and found that FDI in terms of joint ventures had positive direct and spillover effects. However, FDI in a completely foreign-owned firm carried the opposite effects. *Chen et al. (2022)* investigated 69 Chinese cities and found spatial links in carbon productivity. Furthermore, industrial structure reduced carbon productivity. However, political resource endowment increased carbon productivity. *Hu & Wang (2020)* investigated China in spatial analyses and found that following weak to strong regulation shifted a negative effect toward a positive impact of regulation on CP in the direct and spillover effects.

*Long et al. (2020)* studied Chinese provinces from 1998–2016 in spatial analyses and found that FDI increased carbon productivity. However, the spillovers of FDI reduced carbon productivity in neighboring provinces. Moreover, industrial development also played a moderating role in this relationship. *Pan et al. (2020)* examined 30 Chinese provinces from 2004–2016 and reported that outward FDI improved regional TFCP and TFCP of neighboring provinces. *Feng, Shulian & Renjin (2022)* explored China from 2010–2019 and substantiated spillovers in carbon productivity between provinces. Furthermore, fiscal decentralization in terms of revenues and expenditures increased carbon productivity. However, indirect effects were found to be negative in neighboring provinces.

*Hou, Yu & Fei (2023)* explored Chinese provinces from 2011–2019 and environmental regulation had a U-shaped spatial effect on carbon productivity. This U-shaped effect flattened with the moderation of pollution transfers and became steeper with technological progress. *Meng et al. (2023)* investigated 30 Chinese provinces from 2011–2020 and corroborated the spatial links in carbon productivity among provinces. Moreover, REC and innovation helped to raise carbon productivity. *Zhao et al. (2023)* explored 30 Chinese provinces from 1998–2020 and found that environmental R&D had a U-shaped effect on

carbon productivity in China. Moreover, government support for such R&D played a positive role in enhancing carbon productivity. The spatial effects of government support and R&D were also found significant on CP. *Yao, Zhang & Wang (2023)* investigated 281 Chinese cities from 2003–2017 and found that economic agglomeration through technical progress helped to raise the carbon productivity in the cities and the surrounding cities with spatial effects. *Sun, Chen & Wang (2023)* explored 201 Chinese cities from 2011–2020 and found that digital finance through human capital improved carbon productivity in the local and neighboring cities with spatial effects.

The determinants of CP have not been investigated in the GCC region. However, literature has explored the impacts of NRRs, urbanization, trade variables, and FDI on emissions in the GCC region. For instance, *Mahmood & Furqan (2021)* explored GCC countries from 1980–2014 and substantiated that oil rent, energy usage, and urbanization increased $CO_2$ emissions. However, FDI decreased $CO_2$ emissions. Similarly, *Al-Mulali & Tang (2013)* originated that FDI reduced $CO_2$ in the GCC region from 1980–2017. However, energy usage and income growth increased $CO_2$ emissions. *Zmami & Ben-Salha (2020)* explored GCC economies from 1980–2017. FDI increased and urbanization decreased $CO_2$ emissions. *Majeed et al. (2021)* analyzed GCC economies from 1990–2018 and found that globalization, REC, and NRRs helped to improve the environment. However, urbanization, non-REC, and economic growth reduced environmental quality. The GCC literature has investigated the effects of NRRs, urbanization, trade, and FDI on $CO_2$ emissions. However, the determinants of carbon productivity (an indicator of sustainable development) are missing in GCC literature. Thus, this present study fills this gap.

The literature highlights the importance of investigating the determinants of carbon productivity. Particularly, the literature has examined the effects of trade, FDI, and urbanization on CP. However, the role of natural resources in determining CP is ignored in the global and GCC literature. Therefore, the present study explores a relationship between OR and NGR and carbon productivity in six GCC countries. Moreover, spatial dimensions are also considered in these geographically nearby economies.

## METHODS

The present study is motivated to estimate the determinants of carbon productivity (GDP/$CO_2$ emissions), which could affect both GDP and $CO_2$ emissions. For instance, exports and imports are components of GDP and could increase GDP in the case of a surplus trade balance. However, these factors can reduce GDP in the event of a deficit in the trade balance. Thus, trade is an important determinant of GDP and carbon productivity. In addition, trade can be considered a fundamental factor in determining $CO_2$ emissions. For instance, trade can have a scale effect on emissions (*Grossman & Krueger, 1994*). Thus, trade can reduce carbon productivity by increasing $CO_2$ emissions. The scale effect explains that increasing income due to surplus trade would raise economic activities and energy usage, which could pollute the environment. Alternatively, trade can have dominant technique and composition effects. For instance, trade with developed countries might force the economies to follow clean standards (*Birdsall & Wheeler, 1993*;

*Komen, Gerking & Folmer, 1997*), which can develop the technique and composition effects in the industrial sector. For instance, the technique effect of trade explains that trade is a source of technology transfer and can provide the latest clean technology to open economies (*Arrow et al., 1995*). Thus, the trade would increase the use of the latest clean technologies and could have a pleasant environmental technique effect. Moreover, trade may also be forced to follow tight environmental standards, which can change the composition of the industry in favor of clean processes and products. Additionally, FDI can affect both GDP and $CO_2$ emissions. FDI may increase production and can contribute to GDP. Moreover, FDI is a source of technology transfer (*Letchumanan & Kodama, 2000*). Therefore, the latest technology can help reduce $CO_2$ emissions by technique and composition effects. However, FDI may also pollute the environment if FDI enters the dirty industry (*Mani & Wheeler, 1998*), which may result in a dominant scale effect. Thus, FDI may have an impact on $CO_2$ emissions and can determine carbon productivity. Moreover, urbanization can also increase $CO_2$ emissions due to the increasing consumption of energy-intensive products in urban areas (*Shukla & Parikh, 1992*). Natural resource rents could be a blessing by increasing the GDP of resource-rich countries (*Islam, Al Abbasi & Dey, 2023*; *Ben-Salha, Dachraoui & Sebri, 2021*). Alternatively, natural resources can be a curse as explained by Dutch Disease (*Sachs & Warner, 1995*; *Meadows et al., 1972*), and can reduce economic growth in resource-rich countries (*Sha, 2023*; *Ze et al., 2023*; *Safdar, Khan & Andlib, 2022*). On the environmental side, NNRs could add to $CO_2$ emissions (*Peng et al., 2023*; Pu et al., 2013; *Danish, Ulucak & Baloch, 2023*; *Umar et al., 2020*). Resource extraction releases combustion gases, including carbon emissions. Therefore, natural resources could determine both GDP and $CO_2$ emissions and consequently might determine carbon productivity. So, we add oil and natural gas rents to the carbon productivity model of oil and natural gas-rich GCC economies. Based on theoretical discussions, we hypothesize the following model:

$$CP_{it} = f(OR_{it}, NGR_{it}, UP_{it}, TO_{it}, FDI_{it}) \tag{1}$$

$CP_{it}$ is a natural logarithm of carbon productivity, which is a ratio of GDP in dollars to territorial $CO_2$ emissions in tons (*Kaya & Yokobori, 1997*). $OR_{it}$ is a natural logarithm of the oil rent percentage of the GDP. $NGR_{it}$ is a natural logarithm of the natural gas rent percentage of the GDP. $UP_{it}$ is a natural logarithm of the urban population percentage of the total population. $TO_{it}$ is a natural logarithm of the total trade percentage of the GDP, which is a proxy for trade openness. $FDI_{it}$ is the net FDI inflows percentage of the GDP. $FDI_{it}$ is not taken in logarithm as it carries negative values along positive values. *i* represents six GCC countries. *t* displays annual series from 1980–2021. A maximum time sample is used as per data availability. Moreover, the sample of all GCC countries is chosen as all GCC economies are oil and gas producers. Hence, the NRRs could affect both GDP and carbon emissions in the GCC region. The data on territorial $CO_2$ emissions are obtained from *Global Carbon Atlas (2023)* and the rest of all data is sourced from the *World Bank (2023)*.

After discussing the model of the study, we apply the non-spatial fixed effect (FE) and pooled ordinary least square (POLS) testing the statistical existence of spatial

autocorrelation suggested by *Elhorst (2010)*. For this purpose, the Lagrange multiplier (LM) test can be used (*Anselin, Le Gallo & Jayet, 2008*) and the LM robust test can be applied to verify the results of the LM test (*Debarsy & Ertur, 2010*). These tests have good power to examine the possible spatial dependency in a non-spatial model. If the spillovers are corroborated statistically, then the results of non-spatial models are biased (*Elhorst, 2010*). The testing of spatial dimensions is pertinent for nearby countries in a region because trading and ecological policies of the countries from the same region can be responsible for spatial dependence of emissions (*Maddison, 2007*). Moreover, carbon emissions are global emissions. So, the emissions of any country may transfer to the neighboring countries. Keeping in view the possible spatial dimension, many recent studies are utilized spatial econometrics to explore the determinants of carbon productivity (*Yao, Zhang & Wang, 2023*; *Sun, Chen & Wang, 2023*; *Hou, Yu & Fei, 2023*; *Meng et al., 2023*; *Zhao et al., 2023*). In addition, the GCC countries share the same landscape and climate and all GCC countries are oil producers as well. So, emissions of the GCC region could have spatial linkages. In the same way, carbon productivity could have spatial links, which should be tested by statistical tools. If LM and LM robust tests could confirm the statistically significant spillover links, then the spatial Durbin model (SDM) may be utilized using Eq. (2):

$$CP_{it} = \alpha_{10} + \alpha_{11}OR_{it} + \alpha_{12}NGR_{it} + \alpha_{13}UP_{it} + \alpha_{14}TO_{it} + \alpha_{15}FDI_{it} +$$

$$\beta_{11} \sum_{i \neq j, j=1}^{n} W_{ij}OR_{jt} + \beta_{12} \sum_{i \neq j, j=1}^{n} W_{ij}NGR_{jt} + \beta_{13} \sum_{i \neq j, j=1}^{n} W_{ij}UP_{jt} + \beta_{14} \sum_{i \neq j, j=1}^{n} W_{ij}TO_{jt} + \qquad (2)$$

$$\beta_{15} \sum_{i \neq j, j=1}^{n} W_{ij}FDI_{jt} + \delta \sum_{i \neq j, j=1}^{n} W_{ij}CP_{jt} + v_{1i} + u_{1t} + e_{1it}$$

$W_{it}$ is a matrix carrying the distance from country $i$ to neighboring country $j$ in the GCC region in kilometers. Particularly, the inverse distance is utilized in the matrix to reflect more weight for the closed economies and less weight for distanced economies. $W_{it}$ has dimensions of 6 * 6 and is multiplied with all variables to capture the spillover effects. In addition, *Kelejian & Prucha (2010)* advised normalizing this matrix before applying regression. $W_{it}$ is normalized by dividing each element of a row by the sum of the row. Equation (2) is SDM specification and can be tested for its suitability over other spatial specifications, *i.e.*, Spatial Autoregressive (SAR) and Spatial Error Model (SEM). For this purpose, the Wald test can be applied with a null hypothesis $\beta = 0$. If this hypothesis is accepted, SAR can be superior to SDM. Afterward, the Wald test can be applied with a null hypothesis $\beta + \delta.\alpha = 0$. If this hypothesis is accepted, SEM can be superior to SDM. Afterward, the likelihood ratio (LR) test can be used to verify the results of the Wald test. SDM can be considered for the most robust results if both hypotheses are rejected. The unweighted variables in Eq. (2) of SDM can capture the direct effects of independent variables on carbon productivity. Moreover, the weighted variables capture the spillover effects of $OR_{jt}$, $NGR_{jt}$, $UP_{jt}$, $FDI_{jt}$, $TO_{jt}$, and $CP_{jt}$ of country $j$ on the $CP_{it}$ of country $i$. On the other hand, if SAR is superior to other spatial models, then the SAR model can be defined as follows:

$$CP_{it} = \alpha_{20} + \alpha_{21}OR_{it} + \alpha_{22}NGR_{it} + \alpha_{23}UP_{it} + \alpha_{24}TO_{it} + \alpha_{25}FDI_{it} + \delta \sum_{i \neq j, j=1}^{n} W_{ij}CP_{jt} \quad (3)$$

$$+ v_{2i} + u_{2t} + e_{2it}$$

$\delta$ is capturing the spatial effect of carbon productivity from country $j$ to $i$. In case, SEM is superior to other spatial models, the SEM model is presented as follows:

$$CP_{it} = \alpha_{30} + \alpha_{31}OR_{it} + \alpha_{32}NGR_{it} + \alpha_{33}UP_{it} + \alpha_{34}TO_{it} + \alpha_{35}FDI_{it} + v_{3i} + u_{3t} + e_{3it} \quad (4)$$

$$e_{3it} = \sum_{i \neq j, j=1}^{n} W_{ij}\Psi_{it} + \omega_{it} \quad (5)$$

In Eq. (4), the spatial effect is added by Eq. (5) through the weighted error term.

## RESULTS AND DISCUSSION

First, we present the results of non-spatial models in Table 1. POLS estimates show that all coefficients are statistically insignificant. However, we regress the FE models with different specifications and apply the LR tests to verify the FE suitability over POLS. The test statistics from the LR test are 143.20, 342.11, and 532.51 for FE-countries, FE-time, and FE-time and countries, respectively. The null hypothesis (POLS is well-fitted) is rejected in all FE specifications. Thus, the results of the LR test exhibit that all FE models are preferable over POLS. FE results show that oil rent and urbanization have negative effects on carbon productivity. However, NGR and trade openness are enhancing carbon productivity and FDI has an insignificant effect. Then, we apply the LM test to verify the spatial dimensions. It confirms the spatial lag effects in all estimates with test statistics 692.882, 415.737, 246.700, and 215.258. The robust LM test also validates these findings with test statistics 22.199, 167.080, 180.353, and 187.217. Moreover, the spatial error effects are corroborated by LM test statistics 675.213, 248.669, 68.466, and 35.909. The robust LM test also validates these results with test statistics 4.529 and 7.867 in the case of POLS and FE-time and countries, respectively. However, the robust LM test could not validate the spatial error effects in FE-country and FE-time with test statistics of 0.011 and 2.119, respectively. Consequently, the spatial lag effect is dominant. Thus, we apply SDM and SAR specifications for further estimations and ignore the results of non-spatial estimates.

In Table 2, we estimate SDM and utilize the LR and Wald tests to substantiate the consistency of SDM on other spatial models. Both tests reject $\beta = 0$ and $\beta + \delta.\alpha = 0$ with test statistics 47.25 and 47.199 in the case of spatial lag effects and with test statistics 36.45 and 40.18 in the case of spatial error effects. Thus, the SDM is the most appropriate spatial specification. However, we present both SAR and SDM results to display the robustness of the findings. Moreover, we apply the Hausman test for both SAR and SDM specifications. The test statistics are 258.864 and 198.254 in the SDM and SAR models, respectively. Thus, FE is efficient in both SDM and SAR results. The weighted coefficient of carbon productivity is positive in the SDM. It explains that the increasing CP (GDP/$CO_2$ emissions) of one GCC country has spillovers on the CP of neighboring GCC countries as

| Table 1 Non-spatial results. | | | | |
|---|---|---|---|---|
| Variable | POLS | FE-countries | FE-time | FE-time and countries |
| $OR_{it}$ | −0.1133 (0.135) | −0.3755 (0.000) | −0.3647 (0.000) | −0.4541 (0.000) |
| $NGR_{it}$ | 0.0514 (0.147) | 0.2451 (0.000) | 0.2006 (0.000) | 0.1183 (0.013) |
| $UP_{it}$ | −0.3672 (0.158) | −1.0779 (0.033) | −0.4736 (0.002) | −0.7747 (0.015) |
| $TO_{it}$ | 0.0233 (0.849) | 0.7691 (0.000) | 0.2388 (0.002) | 0.2026 (0.030) |
| $FDI_{it}$ | 0.0067 (0.496) | 0.0025 (0.740) | 0.0153 (0.114) | 0.0030 (0.492) |
| Diagnostic tests | | | | |
| LM test-spatial lag | 692.882 (0.000) | 415.737 (0.000) | 246.700 (0.000) | 215.258 (0.000) |
| Robust LM test-spatial lag | 22.199 (0.000) | 167.080 (0.000) | 180.353 (0.000) | 187.217 (0.000) |
| LM test-spatial error | 675.213 (0.000) | 248.669 (0.000) | 68.466 (0.000) | 35.909 (0.000) |
| Robust LM test-spatial error | 4.529 (0.033) | 0.011 (0.915) | 2.119 (0.145) | 7.867 (0.005) |
| $\sigma^2$ | 0.2679 | 0.1549 | 0.0827 | 0.0398 |
| $R^2$ | 0.2370 | 0.4469 | 0.7488 | 0.8519 |
| LR test | | 143.20 (0.000) | 342.11 (0.000) | 532.51 (0.000) |

Note:
() carry $p$-values.

well. Thus, GDP and $CO_2$ emissions have spillovers in neighboring economies. Particularly, increasing emissions in one economy may have environmental spillovers in neighboring countries. Moreover, economic growth may also have spatial links due to some common trading and economic policies in the GCC region. Many spatial studies have substantiated the spillovers of carbon productivity in neighboring economies (*Yao, Zhang & Zheng, 2022*; *Meng, Sun & Guo, 2022*; *Long, Shao & Chen, 2016*; *Zhang et al., 2018*; *Han, 2021*; *Chen et al., 2022*; *Feng, Shulian & Renjin, 2022*; *Yao, Zhang & Wang, 2023*; *Meng et al., 2023*).

$OR_{it}$ has a direct negative impact on $CP_{it}$ in the domestic economies of GCC countries. Oil rent is a significant contributor to the GDP of GCC economies due to the over-dependence on the oil sector. However, the oil sector releases a lot of emissions from the downstream to upstream in the supply chain of oil production. So, the oil sector has environmental concerns for GCC countries. The estimated negative effect corroborates that the impact of oil rent is more dominant on $CO_2$ emissions than on the GDP. This matches the fact that the oil sector could have more environmental concerns compared to other energy or economic sectors in the GCC region. Thus, increasing oil rent has stronger effects on emissions, which are responsible for higher $CO_2$ emissions and lower CP in the domestic economies. However, the indirect impact of oil rent is negative but insignificant. Thus, increasing oil rent is not affecting the carbon productivity in neighboring economies. Nevertheless, the total effect is negative. So, increasing oil rent decreases carbon productivity in the whole GCC region. This result shows that over-reliance on the oil sector has more environmental concerns than the expected positive economic growth effect in the whole region and *vice versa* in case of decreasing oil rent. So, increasing oil rent is reducing carbon productivity, and reducing oil rent could help in increasing carbon productivity in the whole GCC region. *Mahmood & Furqan (2021)* stated a positive effect

| | SDM<br>Coefficient (*p*-value) | SAR<br>Coefficient (*p*-value) |
|---|---|---|
| **Table 2 Spatial results.** | | |
| Point estimates | | |
| $OR_{it}$ | −0.7490 (0.000) | −0.3372 (0.000) |
| $NGR_{it}$ | 0.1190 (0.028) | 0.1274 (0.000) |
| $UP_{it}$ | −1.0794 (0.014) | −1.5108 (0.000) |
| $TO_{it}$ | 0.3537 (0.007) | 0.1531 (0.191) |
| $FDI_{it}$ | 0.0104 (0.031) | 0.0003 (0.957) |
| Direct estimates | | |
| $OR_{it}$ | −0.7035 (0.000) | −0.4318 (0.000) |
| $NGR_{it}$ | 0.0294 (0.498) | 0.1619 (0.000) |
| $UP_{it}$ | −0.7477 (0.033) | −1.8971 (0.000) |
| $TO_{it}$ | 0.1915 (0.104) | 0.1919 (0.189) |
| $FDI_{it}$ | 0.0048 (0.210) | 0.0004 (0.956) |
| Indirect estimates | | |
| $OR_{it}$ | −0.3063 (0.370) | −0.6732 (0.000) |
| $NGR_{it}$ | 0.6274 (0.001) | 0.2510 (0.000) |
| $UP_{it}$ | −1.9820 (0.021) | −2.9660 (0.001) |
| $TO_{it}$ | 1.1239 (0.006) | 0.2881 (0.198) |
| $FDI_{it}$ | 0.0395 (0.010) | 0.0005 (0.964) |
| Total estimates | | |
| $OR_{it}$ | −1.0098 (0.013) | −1.1050 (0.000) |
| $NGR_{it}$ | 0.6567 (0.002) | 0.4129 (0.000) |
| $UP_{it}$ | −2.7296 (0.013) | −4.8632 (0.192) |
| $TO_{it}$ | 1.3154 (0.003) | 0.4800 (0.192) |
| $FDI_{it}$ | 0.0442 (0.009) | 0.0009 (0.961) |
| Weights | | |
| $W*OR_{it}$ | −0.8903 (0.073) | |
| $W*NGR_{it}$ | 0.9275 (0.000) | |
| $W*UP_{it}$ | −3.4183 (0.016) | |
| $W*TO_{it}$ | 1.7097 (0.005) | |
| $W*FDI_{it}$ | 0.0598 (0.009) | |
| $W*CP_{it}$ | 0.6019 (0.009) | 0.6903 (0.000) |
| Diagnostic tests | | |
| $R^2$ | 0.4569 | 0.2193 |
| $\sigma^2$ | 0.0205 (0.000) | 0.0742 (0.000) |
| Hausman test | 258.864 (0.000) | 198.254 (0.000) |
| Spatial lag-LR test | 47.25 (0.000) | |
| Spatial lag-Wald test | 47.19 (0.000) | |
| Spatial error-LR test | 36.45 (0.000) | |
| Spatial error-Wald test | 40.18 (0.000) | |

of oil rent on $CO_2$ emissions in GCC. Accordingly, oil rent can reduce carbon productivity and our results have also substantiated their findings. However, *Majeed et al. (2021)* validated a negative effect of NRRs on $CO_2$ emissions in the GCC region. Their results may show a dominance of NGR in total NRRs. However, this present research has tested the separate effects of oil and gas rents, which would help to understand the environmental effects of both OR and NGR separately.

$NGR_{it}$ has a positive and insignificant effect in direct estimates. Thus, natural gas rent could not affect carbon productivity in domestic economies. This may be claimed due to a limited proportion of natural gas rent in the GDP of most GCC economies. Nevertheless, the indirect effect of $NGR_{it}$ is positive and significant. Thus, increasing natural gas rent in one GCC economy could raise the carbon productivity in the neighboring economies. Moreover, $NGR_{it}$ has a positive effect in the total effect and it helps to raise the CP in the whole GCC region. The natural gas sector is less polluted compared to the oil sector and releases fewer emissions compared to the oil sector. Moreover, the natural gas sector could have a pleasant growth effect, which is greater than the growth of emissions. Thus, the net total effect of $NGR_{it}$ is positive on carbon productivity in the whole GCC region. So, natural gas rent helps to achieve sustainable economic growth in the region. *Peng et al. (2023)* also verified our results by reporting the negative effect of NGR and the positive effects of coal and mineral rents on $CO_2$ emissions. Thus, natural gas production is less polluted than other natural resources production *i.e.*, oil, coal, and other mineral resources.

Urbanization reduces carbon productivity in all estimates. Hence, urbanization is reducing carbon productivity in domestic and neighboring economies. Moreover, it is responsible for reducing carbon productivity in the whole GCC region as well. GCC economies are highly urbanized and use heavy vehicles and other energy-intensive products in urban areas. Moreover, the warm climate of the GCC region needs more energy for cooling purposes in the most of seasons. In addition, GCC countries are high-income economies and use fossil fuels intensively due to low local energy prices. Thus, urbanization is responsible for higher $CO_2$ emissions and lower carbon productivity in the GCC region. Likewise, *Majeed et al. (2021)* substantiated a positive impact of urbanization on $CO_2$ emissions in the GCC region. Thus, urbanization could reduce CP by increasing $CO_2$ emissions. Moreover, some non-GCC studies also investigated and found a negative effect of urbanization on CP (*Wang, Li & Li, 2022*; *Murshed et al., 2022*). Thus, urbanization is also responsible for environmental degradation in non-oil producer countries.

$TO_{it}$ carries a positive and insignificant effect on $CP_{it}$ in direct effect. However, trade openness carries a positive impact in indirect and total estimates. *Zhang et al. (2018)* also reported the positive spillovers of trade on CP. Trade openness helps increase GDP and/or reduce $CO_2$ emissions in neighboring countries and the entire region. For instance, the balance of trade (exports minus imports) remains surplus in most of the sample years in GCC countries. Thus, surplus trade is increasing GDP. Moreover, if trade openness increases emissions less than its effect on GDP or reduces emissions, then the ratio of carbon productivity (GDP/$CO_2$ emissions) may rise. On the whole, trade openness raises sustainable growth in the GCC region. Thus, the results of trade openness show that trade

has dominant positive environmental technical and composition effects compared to the possible negative environmental scale effects. In the same way, *Li & Wang (2019)* also reported a positive effect of TO on CP in China. However, *Bai et al. (2019)* found a positive effect of TO on CP in the case of 88 economies. This result may be due to the reason that most of their sample countries have a deficit in trade balance, which is responsible for reducing GDP.

$FDI_{it}$ has a positive and insignificant impact in direct estimates and carries a positive impact in indirect and aggregate estimates. Similarly, *Tang et al. (2022)* found positive spillovers of FDI on carbon productivity. Conversely, *Long et al. (2020)* found negative spillovers of FDI. FDI could help in raising carbon productivity in neighboring GCC countries by reducing $CO_2$ emissions. Our results also demonstrate that FDI in one GCC country helps to raise the carbon productivity of neighboring GCC countries. Moreover, FDI also helps to raise carbon productivity in the total estimates. It may be claimed due a reason that foreign investors generally employ better technology than local investors, which would create technique effects (*Letchumanan & Kodama, 2000*). Moreover, foreign investments would also flow into the cleaner sectors, which would have a composition effect. Along the same lines, past studies also reported a positive effect of FDI on carbon productivity (*Zhang & Xu, 2016*; *Tang et al., 2022*; *Long et al., 2020*). In GCC literature, studies substantiated a negative impact of FDI on $CO_2$ emissions in the GCC region (*Mahmood & Furqan, 2021*; *Al-Mulali & Tang, 2013*). Therefore, FDI could also have a positive impact on CP, which is substantiated by the results of the present study. On the whole, both indicators of globalization, *i.e.*, FDI and TO are helping to raise carbon productivity in the GCC region.

## CONCLUSION

Globalization, urbanization, OR, and NGR could determine the carbon productivity in resource-rich countries. Therefore, we explore the effects of trade openness, FDI, urbanization, OR, and NGR on CP in the GCC region caring about the spatial dimensions in analyses. The past GCC literature has worked on the role of some of the investigated variables on carbon emissions. However, this present research contributes to the GCC literature by exploring the determinant of carbon productivity, which is a strong indicator of sustainable development. For this purpose, the present study uses the period of 1980–2021 and the SDM for data analyses. We find the positive spillovers of carbon productivity of one GCC country to neighboring GCC countries. Thus, increasing carbon productivity in one country helps raise carbon productivity in neighboring GCC economies. Oil rent is reducing the carbon productivity in domestic economies but the spillover effect is found insignificant. Moreover, oil rent is reducing carbon productivity in the whole GCC region. Natural gas rent has an insignificant effect on CP in domestic economies and has positive spillovers in neighboring GCC countries. Further, natural gas rent helps in raising carbon productivity in the whole GCC region. Thus, natural gas rent carries pleasant environmental and economic growth effects in the GCC region. Urbanization decreases carbon productivity in domestic and neighboring countries and the entire region. So, increasing urbanization in the GCC countries has environmental concerns for the whole

region. Trade openness and FDI could not affect carbon productivity in domestic economies but are helping to raise CP in neighboring economies with a spillover effect and in the whole GCC region. Thus, the technique and composition effects of FDI and trade openness are dominant over the scale effects. Thus, both indicators of globalization helped to raise the GDP of the region.

Oil rent is reducing carbon productivity in the region. Therefore, the GCC region is suggested to reduce reliance on oil rent. It can be achieved by diversifying the GCC economies from the oil sector to non-oil and less-pollution-oriented products. Further, natural gas rent is increasing carbon productivity in the entire GCC region. Thus, the reliance on natural gas rent should be enhanced to increase CP. Urbanization decreases CP in all estimates. Thus, the government should impose a pollution tax on energy-intensive urban products. FDI is increasing CP in the entire GCC region. The GCC countries should relax the taxes and provide financial incentives to attract foreign investors to promote CP in the region. In addition, trade openness should also be encouraged as it has a positive effect on CP. Particularly, non-oil and less-pollution-oriented exports should be encouraged. Increasing these exports will generate more trade surplus, which will contribute GDP of GCC economies on one hand. On the other hand, these exports will reduce aggregate carbon emissions in the region. As a result, these exports would help raise carbon productivity.

The present research could work on the countries in the GCC region. However, future research may enhance the geographical sample by adding other MENA region countries to increase the scope of the results. Moreover, technology variables can also be added in future research to observe the effect of technological development on carbon productivity.

### Funding
This work was supported by the Deputyship for Research & Innovation, Ministry of Education in Saudi Arabia through project number (IF2/PSAU/2022/02/21824).
The funders had no role in study design, data collection and analysis, decision to publish, or preparation of the manuscript.

### Grant Disclosures
The following grant information was disclosed by the authors:
Deputyship for Research & Innovation.
Ministry of Education in Saudi Arabia: IF2/PSAU/2022/02/21824.

### Competing Interests
Haider Mahmood is an Academic Editor for PeerJ.

### Author Contributions
- Haider Mahmood conceived and designed the experiments, performed the experiments, analyzed the data, prepared figures and/or tables, authored or reviewed drafts of the article, and approved the final draft.

## Data Availability

The raw data are available in the Supplemental File.

## Supplemental Information

Supplemental information for this article can be found online at http://dx.doi.org/10.7717/peerj.16281#supplemental-information.

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
