# Peer review of "Spatial effects of trade, foreign direct investment (FDI), and natural resource rents on carbon productivity in the GCC region"

_PeerJ, doi:10.7717/peerj.16281_

## Round 0.1 · original submission · Major Revisions

Authors are advised to revise the manuscript as suggested by reviewers.

Reviewer 1 ·

Basic reporting

The background of the abstract section should be enhanced.
The methods of the abstract section should also add discussions of particular spatial technique.
The abstract section should also be enhanced in terms of policy discussions.
Keywords should be added after the abstract section.
In line 31, carbon productivity should be capitalized to define the abbreviation of CP.
The statement in lines 44-46 needs more clarity to understand the contribution of surplus trade to economic growth.
The last two paragraphs of the introduction section should further focus on the literature gap and the scientific contribution to the literature.
The literature review is well written. However, the latest studies published in 2023 should be added. I have seen some latest studies in theoretical discussions in the methods section. However, the literature review section also needs attention in this regard.
The first paragraph of the methods section should define the scale, composition, and technique effects. Then, the effect could be linked with the discussed variables of the study.
The discussions of major results in the conclusion section must be increased.

Experimental design

It is advised to define the weight matrix with more details.
The components of equation 2 can be explained in more detail.
The choice of technique can be discussed in more detail.
More explanations are needed about the sample.
The diagnostic tests can be discussed in more detail.

Validity of the findings

The findings of the study are well-presented. However, the interpretation of the results should be enhanced. Particularly, the discussions on the comparisons of the results with past literature should be increased.

Additional comments

The conclusion section is short and the discussions of the results in this section should be increased. Moreover, the discussions on policy implications should be increased.

Reviewer 2 ·

Basic reporting

Basic Reporting: The article “Spatial effects of trade, FDI, and natural resource rents on carbon productivity in the GCC region” has investigated an interesting topic for an interesting case of GCC countries. The study found that oil rent and urbanization reduce carbon productivity and natural gas increases carbon productivity. Both natural resource rents also have spillovers in neighboring countries. Moreover, trade openness and foreign investments helped in increasing carbon productivity with spillover effects. However, I suggest the following recommendations to improve the quality of the research:
Introduction: The first paragraph of the introduction section explains the importance of the investigation of carbon productivity over carbon emissions. However, carbon productivity should be defined first before providing any arguments. The third paragraph of the introduction section should further discuss the role of foreign investments in determining carbon productivity in terms of both income and carbon emissions. Dutch Disease has been discussed in the introduction section, which needs further explanation of how Dutch Disease may obstruct the growth of other economic sectors and activities in an economy.
Literature review section: The statement “CP is pertinent for the sustainable growth of any country” needs reference. Song & Han (2022) found that ecological standards increased carbon productivity. The channels for such a relationship should be discussed in more detail. In the same way, the study of Li & Liu (2022) should be discussed in more detail. The statement “Carbon emissions are global emissions” needs reference. Long, Shao & Chen (2016) found a significant spatial correlation in carbon productivity. This statement needs an explanation to understand the technical meaning of spatial correlation. The comments for references Meng, Sun & Guo (2022) and Han (2021). The second last paragraph of the literature review needs closing lines to explain the literature gap in the context of GCC countries and how the present study might fill this gap.

Experimental design

The method section is well-described. However, there is a need to add more justifications for using the spatial approach instead of other standard econometric techniques. The SAR and SEM equation should also be written in this section as SDM has a tendency to reduce to both specifications and these are regressed to verify the appropriateness of the SDM. The normalizing procedure of Kelejian & Prucha (2010) should be further explained.

Validity of the findings

Validity of the findings: The validity of the findings is tested in two layers. First, the pooled OLS and fixed effects regressions are tested for the existence of spatial dimensions. Moreover, the SDM is compared with the SAR and SEM by using standard diagnostic tests. These all testing needs statistical values to support the acceptance and the rejection of the hypotheses. So, the readers may understand how all procedures helped to approach the most robust results. In addition, the discussions should be enhanced to explain the economic meanings of results.

Additional comments

Additional comments:
The conclusion section is well-written. I suggest adding the net contribution of this research to the existing literature. Moreover, limitations and the future direction can be added to increase the scope of the research.

---

## Round 0.2 · accepted · Accept

The authors have revised the manuscript as per the suggestions of the reviewers. Therefore, I recommend that the manuscript be accepted for publication. Regards

Reviewer 1 ·

Basic reporting

No further comments

Experimental design

No further comments

Validity of the findings

No further comments

Additional comments

No further comments

Reviewer 2 ·

Basic reporting

The author addressed all the comments in a satisfactory way. I have no further suggestions,

Experimental design

experimental design seems up to the mark.

Validity of the findings

The conclusion and findings are interlinked and i have no further suggestions for the authors.

Additional comments

I am, pleased to recommend the acceptance of the manuscript.